# Differential Growth Characteristics of Crimean-Congo Hemorrhagic Fever Virus in Kidney Cells of Human and Bovine Origin

**DOI:** 10.3390/v12060685

**Published:** 2020-06-25

**Authors:** Katalin Földes, Touraj Aligholipour Farzani, Koray Ergünay, Aykut Ozkul

**Affiliations:** 1Virology Department, Faculty of Veterinary Medicine, Ankara University, Ankara 06110, Turkey; fldeskatalin@gmail.com (K.F.); touraj.farzani@gmail.com (T.A.F.); 2Division of Infectious Diseases and International Medicine (IDIM), University of Minnesota, Twin Cities, MN 55455, USA; 3Virology Unit, Department of Medical Microbiology, Faculty of Medicine, Hacettepe University, Ankara 06100, Turkey; ekoray@hacettepe.edu.tr; 4Biotechnology Institute, Ankara University, Ankara 06560, Turkey

**Keywords:** Crimean-Congo hemorrhagic fever virus, in vitro characterization, cytotoxicity, immunofluorescence, virus productivity

## Abstract

Crimean-Congo hemorrhagic fever virus (CCHFV) causes a lethal tick-borne zoonotic disease with severe clinical manifestation in humans but does not produce symptomatic disease in wild or domestic animals. The factors contributing to differential outcomes of infection between species are not yet understood. Since CCHFV is known to have tropism to kidney tissue and cattle play an important role as an amplifying host for CCHFV, in this study, we assessed in vitro cell susceptibility to CCHFV infection in immortalized and primary kidney and adrenal gland cell lines of human and bovine origin. Based on our indirect fluorescent focus assay (IFFA), we suggest a cell-to-cell CCHF viral spread process in bovine kidney cells but not in human cells. Over the course of seven days post-infection (dpi), infected bovine kidney cells are found in restricted islet-like areas. In contrast, three dpi infected human kidney or adrenal cells were noted in areas distant from one another yet progressed to up to 100% infection of the monolayer. Pronounced CCHFV replication, measured by quantitative real-time RT-PCR (qRT-PCR) of both intra- and extracellular viral RNA, was documented only in human kidney cells, supporting restrictive infection in cells of bovine origin. To further investigate the differences, lactate dehydrogenase activity and cytopathic effects were measured at different time points in all mentioned cells. In vitro assays indicated that CCHFV infection affects human and bovine kidney cells differently, where human cell lines seem to be markedly permissive. This is the initial reporting of CCHFV susceptibility and replication patterns in bovine cells and the first report to compare human and animal cell permissiveness in vitro. Further investigations will help to understand the impact of different cell types of various origins on the virus–host interaction.

## 1. Introduction

Crimean-Congo hemorrhagic fever virus (CCHFV) causes a tick-borne viral disease with a geographical distribution in certain endemic areas such as eastern and southern Europe, Asia, the Middle-East, and Africa [1]. It possesses a high mortality rate (up to 40%), and there is no licensed vaccine available to combat the disease [2]. Furthermore, virus behavior and replication characteristics are difficult to study due to the requirement of high containment laboratories. The virus has a high potential for emergence and introduction in new areas and remains a high health risk worldwide [3,4]. 

The pathogenesis and outcome of CCHFV, classified in the Orthonairovirus genus of the Nairoviridae order, vary between human and other mammalian hosts. Ticks serve as a viral vector and reservoir, however many animals, such as cattle have been shown to act as natural hosts for CCHFV, while infected humans do not [5,6,7]. Cattle have an important role in the zoonotic cycle of CCHFV [8]: agricultural practices involving cattle facilitate human-tick contact; slaughterhouses provide an opportunity for direct contact with the viremic blood of infected animals; particular religious practices involving animal sacrifice can contribute to human exposure risks in endemic regions [9]. 

Except for immunocompromised, newborn, or humanized mice and cynomolgus macaques, wild and domestic animals usually develop a short period of viremia with limited or absent clinical signs [5,10,11,12]. However, human CCHFV infections initially produce non-specific symptoms such as fever, anorexia, and diarrhea, which may progress to severe symptoms including petechiae, myalgias, thrombocytopenia, hemorrhage, and eventually death [10,13,14,15]. Little is known about how human and bovine, or other animal cells, differentially respond to CCHFV infection and how replication patterns are associated with disease outcome. Due to limitations in the availability and use of animal models, in vitro cell lines provide a viable approach to model and characterize virus replication processes in different species.

Kidney and adrenal gland cell lines originating from humans and primates such as human adrenal gland/cortex (SW-13), African green monkey kidney (Vero E6), and Rhesus macaque kidney cell lines (LLC-MK2) are frequently used for CCHFV experiments in vitro [16,17,18,19]. In vivo, kidneys are also affected during CCHFV infections in humans, with evidence for direct virally-induced damage in acute renal failure [20,21]. Furthermore, renal failure is usually related to higher mortality rates in human infections [22]. 

There are many unexplained aspects of CCHFV pathogenesis. The strategy of how the virus spreads in the cells and the importance of the innate immune system has been shown to affect human and animal cell line susceptibility and therefore, pathogenesis [23,24,25]. However, comparative cellular responses in human, bovine, or other animal cells during CCHFV remain mostly unexplored. In this study, we investigated in vitro CCHFV replication in immortalized and primary human and bovine kidney cells, for a comparative assessment of cytopathic effects, viral load, cell cytotoxicity, and replication efficiency. 

## 2. Materials and Methods

### 2.1. Ethical Statement

All in vitro biological experiments including the infectious virus manipulation in the cell culture systems were performed in the Biosafety Level 3 plus (BSL3+) facilities of the Virology Department, Veterinary Faculty, Ankara University, Turkey. 

### 2.2. Cells and Culture Conditions

The three human cell lines used in this study included human adrenal gland/cortex Scott and White No. 13 (SW-13), human embryonic kidney (HEK-293), and human primary mesangial cell (HMC). The bovine cells were Madin–Darby bovine kidney (MDBK), bovine embryonic kidney (BEK), and primary bovine kidney cortex cell (PBK) (Table 1). To prepare the primary bovine kidney cell line, the kidney of a five-month-old bull was aseptically removed from the body immediately after the slaughter in a local private abattoir (Cubuk Venture Integrated Meat and Food Industry, Ankara, Turkey) and transferred to the laboratory under chilled conditions. The capsule and medulla were separated from the cortical tissue and the samples were smashed and cut into small pieces. Subsequently, the tissue samples were washed three times using 1x phosphate-buffered saline (PBS; Gibco, Thermo Fisher Scientific, Waltham, MA, USA) and incubated at 37 °C for 45 min with gentle stirring in a papain enzyme solution (1 mg of L-cysteine, 1 mg of BSA, 25 mg of Glucose, 0.5 mg of papain plus 10 µg of DNAse I in 5 mL of 1× PBS). After centrifugation at 300 × *g* for 10 min, the cellular debris was re-suspended in culture medium and cells were cultivated in collagen-coated T25 flasks [26]. The primary bovine cells had three passages before CCHFV infection. MDBK, BEK, and HEK-293 cells were obtained from the departmental culture collection. SW-13 cells were kindly provided by Bernadett Pályi, National Public Health and Medical Officer Service, Hungary, and HMC cells were kindly provided by Prof. Seza Özen Hacettepe University, Ankara, Turkey. The bovine cell lines and HMC were cultured in Eagle’s minimum essential medium (EMEM; Sigma, St. Louis, MO, USA). HEK-293 and SW-13 cells were maintained in minimum essential medium alpha (Thermo Fisher Scientific, Waltham, MA, USA) and Leibovitz’s L-15 medium (Thermo Fisher Scientific, Waltham, MA, USA), respectively. All the media were supplemented with 10% fetal bovine serum (FBS; Biological Industries, Kibbutz Beit-Haemek, Israel), 2 mM L-glutamine (Biological Industries, Kibbutz Beit-Haemek, Israel), 100 U penicillin, and 0.1 mg/mL streptomycin (Thermo Fisher Scientific, Waltham, MA, USA) (Table 1). All cell lines were tested for Mycoplasma contamination by using the EZ-PCR Mycoplasma Test Kit (Biological Industries, Kibbutz Beit-Haemek, Israel) and were sub-cultured in a ratio of 1:2 to 1:4 twice a week. 

### 2.3. Virus and Analysis of Growth Kinetics in the Cell Lines

To investigate viral load and replication pattern in the different human and bovine kidney cell lines, a local CCHFV strain Ank-2 (GenBank Accession number: MK309333) [27] was used in the study. CCHFV strain Ank-2 was isolated in SW-13 cells and after two rounds of virus propagation, virus replication was verified by real-time PCR targeting the S segment and follow-up sequencing. This virus titrated by fluorescence assay was stored at −80 °C as the virus stock for further use. For all kinetic assays, CCHFV Ank-2 strain was used at a 0.1 multiplicity of infection (MOI). To measure the viral RNA load, the cells grown in 24-well plates were infected with the Ank-2 strain in triplicates. After adsorption for 1 h, the non-attached viruses were washed away with PBS and the infected cells were further cultivated for up to seven days. Total RNA extraction from the infected cells and from the supernatant was performed at 0, 1, 2, 3, 5, and 7 days post-infection (dpi). RNA isolation from day 0 samples was performed with 1 h virus adsorption followed by washing steps. For the intracellular S segment analyses, total RNA extraction from the infected cells was performed by the EZ-RNA Total RNA Isolation Kit (Biological Industries, Cromwell, CT, USA) and for the extracellular analyses, RNA from supernatants was taken using the QIAamp Viral RNA Mini Kit (QIAGEN, Germantown, MD, USA) as described by the manufacturer’s instructions. The presence and progress of cytopathic effects (CPEs) were examined on days 1, 2, 3, 5, and 7 post-inoculation using an inverted light microscope (Nikon, Eclipse TS-100, Tokyo, Japan). The mean percentage of the infected cells was used to estimate cellular pathology scores. Cytopathic effect of Ank-2 strain included cell deformation, cell granulation leading to cell detachment, and cell death at the final step. The scoring system was based on assigning numbers (from 0 to 4) to various levels of CPE where 0 indicates no visible change, 1 indicates CPE up to one-quarter (1%–25%), 2 up to half (26%–50%), 3 up to three-quarters, (51%–75%) and 4 up to all (76%–100%) cells of the monolayers. Uninoculated cells were used as negative controls. Virus replication efficiency was tested in parallel by three serial passages in all aforementioned cell lines. For this purpose, each cell line was inoculated in T25 flasks in triplicates at a confluency of 90% up to twelve days until cytopathic effect reached about 80%. After three rounds of freeze-thaw, the cellular debris was removed by centrifugation at 3000 rpm for 10 min at 4 °C and the supernatants were collected. This procedure was repeated two more times. The virus was titrated between each passage using a standard TCID50 assay in SW-13 cells [28]. RNA extraction from the virus-infected aqueous phase of the cell cultures was performed by QIAamp Viral RNA Mini Kit (QIAGEN, Germantown, MD, USA) according to the manufacturer’s instructions. The extracted RNAs were stored at −80 °C.

### 2.4. Quantitative Real-Time Polymerase Chain Reaction (qRT-PCR)

To quantify the amount of the intracellular viral genomic RNA (gRNA) harvested at different time points of incubation, RT-qPCR was conducted by QuantiNova^®^Pathogen+IC Kit (QIAGEN, Germantown, MD, USA). Briefly, the master mix of 15 μL contained 5 μL of 4× Reaction Mix, 6.3 μL of RNase-free water, 1.6 μL of each of forward and reverse primers (0.8 μM final concentration), and 0.5 μL of probe (0.25 μM final concentration). Five microliters of template RNA (500 ng) was added to complete the master mix to 20 μl. The cycling conditions were as follows: cDNA synthesis at 50 °C for 10 min, initial denaturation at 95 °C for 2 min, and 40 cycles of denaturation at 95 °C for 5 s, and annealing/extension at 60 °C for 30 s. The primers and probe were targeting the nucleoprotein gene (small segment) as previously described [27]: qPCR-F (5′-GGACATAGGTTTCCGTGTCA-3′), qPCR-R (5′-TCCTTCTAATCATGTCTGACAGC-3′) and qPCR-probe (5′-FAM-AGAACAACTTGCCAATTACCAACAGGC-BHQ1-3′). Standard linear plasmids were prepared by 10-fold dilutions equivalent to 2 × 10^3^ to 2 × 10^8^ copies/reaction mixture using pCD-N1 as the template [28]. The reactions were carried out in one step by a Rotor-Gene Q instrument (QIAGEN, Germantown, MD, USA) and all calculations were then transformed to log-based viral loads (copies/mL).

### 2.5. Indirect Immunofluorescence Assay (IIFA)

The cells were plated in 24-well plates in triplicates and incubated for 1 day in a 37 °C incubator. 0.1 MOI of the virus was used for inoculation. Subsequently, the infected cells were fixed with 3.7% formaldehyde (Merck KGaA, Darmstadt, Germany) at 1, 2, 3, 5, and 7 dpi and permeabilized by 1% Triton-X-100 (Sigma, St. Louis, MO, USA). After blocking by 5% skimmed milk (Cell Signaling Technologies, Inc., Danvers, MA, USA) for 1 h at room temperature (RT), polyclonal human antiserum was used as a primary antibody in 1:250 dilution. After 1 h 30 min incubation in RT with gentle shaking, fluorochrome-conjugated secondary antibody (Protein G, Alexa Fluor™ 488 conjugate, Invitrogen™, Waltham, MA, USA) was added to the fixed cells at a dilution of 1:1000 and incubated at RT for 1 h. The cells were visualized by an Axio Vert A1 Microscope (Zeiss, Oberkochen, Germany). The mean percentage of positive cells with green fluorescence emission in three 24-wells was used to calculate immunofluorescence scores. The grading was based on a 1 to 4 scoring system where 1 indicates up to one-quarter (1%–25%), 2 indicates up to half (26%–50%), 3 up to three-quarters (51%–75%), and 4 up to almost all (76%–100%) cells emitting fluorescence [29,30]. Uninoculated cells were used as the negative control. To calculate the MOI of the virus in each cell line first, we determined the live cell numbers in a 24-well by Beckman Vi-Cell Cell Viability Analyzer. Later we counted the average of three immunofluorescence-dyed cell numbers/well by Cellcounter (https://bitbucket.org/linora/cellcounter/src/master/) [31]. Pictures used for immunofluorescence-dyed cell count were made on 20× magnification. MOI was counted by the following Equation (1):(1)$$\mathrm{MOI}=\frac{\mathrm{Average}\;\mathrm{fluorescently}\;\mathrm{stained}\;\mathrm{cell}/\mathrm{area}\;\times \;20\;\mathrm{area}/\mathrm{well}\times \;\mathrm{dilution}}{\mathrm{Total}\;\mathrm{cell}\;\mathrm{number}/\mathrm{well}}$$

### 2.6. LDH Assay

To measure lactate dehydrogenase (LDH) levels of CCHFV-infected in the cells at the first three days of infection, we used the Pierce™ LDH Cytotoxicity Assay Kit (Thermo Scientific™, Waltham, MA, USA) as described by the manufacturer’s instructions. Briefly, the cells were plated in 96-well tissue culture plates within 100 µL of the cell-specific medium in triplicate. Cells were incubated in triplicate for water control (Spontaneous LDH Activity Controls) and Lysis Buffer control (Maximum LDH Activity Controls) as well. The next day after plating the cells, 0.1 MOI of virus diluted in 10 µL were inoculated into one set of triplicate wells in each cell culture and 10 µL of molecular-grade water was added to the Spontaneous LDH Activity Controls. Subsequently, the cells were incubated for 1, 2, or 3 days. On the day of analysis, 10 µL of Lysis Buffer was added to Maximum LDH Activity Control wells. After the incubation step and transferring 50 µL of each sample medium to a new 96-well plate, 50 µL of the reaction mixture containing the Assay Buffer and Substrate Mix was added to each well, which was followed by another incubation step and finally 50 µL of stop solution was added to all wells. The absorbance was measured at 490 nm and 680 nm in an ELISA reader (Titertek Multiskan PLUS MK II Microplate Reader, Midland, ON, Canada). The following formula was used to calculate the % cytotoxicity Equation (2):(2)$$\%\;\mathrm{Cytotoxicity}=\frac{\mathrm{Virus}\text{-}\mathrm{treated}\;\mathrm{LDH}\;\mathrm{activity}\;-\;\mathrm{Spontaneous}\;\mathrm{LDH}\;\mathrm{activity}}{\mathrm{Maximum}\;\mathrm{LDH}\;\mathrm{activity}\;-\;\mathrm{Spontaneous}\;\mathrm{LDH}\;\mathrm{activity}}\;\times \;100$$

### 2.7. Statistical Analysis

A two-way ANOVA test (Dunnette’s multiple comparison) was used to compare the mean viral loads in different cell lines at 1, 2, 3, 5, and 7 dpi with the mean viral load at 0 dpi and Sidak’s multiple comparison test was used to compare the mean intra- and extracellular gRNA load of each cell by GraphPad Prism version 6.0 (GraphPad Software, San Diego, CA, USA). Two-way ANOVA (Tukey’s multiple comparision test) was used to compare the LDH levels. All graphs were created by the same software. Calculations of intra- and extracellular genome load were log-transformed (copies/mL). A *p*-value < 0.05 was considered as statistically significant. Standard deviation (SD) and mean were based on triplicate measurements.

## 3. Results

### 3.1. Bovine Kidney Cells Display Islet-Like Infection Pattern while Human Kidney Cells Show Overall Virus Infection and More Pronounced CPE

CCHFV-related CPEs were observed in all of the analyzed cell lines with different cell line-associated patterns (Figure 1 and Figure 2, Table A1). SW-13 cells showed the most prominent CPE including cell rounding at one dpi, which reached >51%–75% of the cells on day three post-infection and by five dpi, most of the cells were detached, rounded, and cell debris was present in the supernatant (Figure 1 and Figure 2B). The virus-induced CPE in HEK-293 cells started at two dpi and was characterized by overall degeneration with rounding of the cells. Most of the HEK-293 cells were affected by the virus at seven dpi (Figure 1 and Figure 2C). The primary human mesangial cell line (HMC) displayed floating detached cell clumps that covered most of the monolayer by the end of seven dpi (Figure 1 and Figure 2A). In comparison to human cells, bovine cells had a delayed onset of CPE and limited pathogenic effect. In all bovine kidney cells (MDBK, BEK, and PBK), CPE started to develop on three dpi and most of the cells remained intact in the monolayer by the end of seven dpi. Less than 50% of the cell monolayer was infected in these cell lines. The immortalized MDBK and BEK bovine kidney cells demonstrated localized areas of cell degeneration and in the primary bovine PBK cells, the elongated morphology became more pronounced and some of the cells lost their barrier integrity (Figure 1 and Figure 2D–F). When we compared the average of the three human and three bovine cells CPE scores, a significant difference was shown from two dpi and onward (*****p*-value < 0.0001) (Figure 2G).

Viral proteins were detected by IIFA in all infected cell lines from the first day after virus inoculation (Figure 2 and Figure 3, Table A1). However, human and bovine cells showed a major difference in the fluorescent cell number and in the infection pattern as infection progressed. In SW-13 cells, infection progressed to >76%–100% from five dpi (Figure 2B and Figure 3B). On three dpi, instead of staying in localized fluorescent plaques (foci), viral proteins were present in up to 75% of HEK-293 cells (Figure 2C and Figure 3C). The HMC primary human cell line demonstrated a slower infection, but the infection rate increased to >76%–100% by seven dpi (Figure 2A and Figure 3A). In all human kidney cells, a generic pattern of infection, characterized by overall spread through the monolayer was observed from three dpi and onward. In contrast, the virus replication in bovine kidney cells remained restricted in specific localized areas and covered less than 50% of the monolayer by the end of the seven dpi (Figure 2 and Figure 3). The primary PBK bovine cell culture displayed smaller groups of infected cells in contact with each other (Figure 3D). The pattern of the infection in immortalized MDBK and BEK bovine cell lines showed big islet-like formations starting from five dpi and the viral particles could be detected only in these islets by the end of seven dpi (Figure 3E,F). Comparing the average of the three human and three bovine IIFA scores, a significant difference was displayed from day three p.i (*****p*-values < 0.0001, Figure 2H). Overall, scoring by CPE and IIFA were consistent with each other.

We analyzed the multiplicity of infection in each cell line by counting fluorescent cells and the total number of the cells in triplicate wells of 24-well plates. Due to the distinct morphology of primary and immortalized cell lines (fibroblastic vs epithelial, respectively), total cell numbers were dependent on cell type. On average, there were 16,000 HMC cells/well and 10,000 PBK cells/well. For immortalized cells, we found on average 35,000 SW-13 cells/well, 30,000 HEK-293 cells/well, and 150,000 BEK and MDBK cells/well. After counting the fluorescently stained cells with the Cellcounter, we obtained prominently different MOIs in human and bovine cells (Figure 2, Table A1). In human cells, MOI was around 1.0 in both SW-13 and HMC (SW-13, 0.90 at five dpi; HMC, 0.97 at seven dpi, Figure 2A,B). Since the complete monolayer was infected by five dpi, SW-13 displayed a decrease in fluorescent cell number at seven dpi. During the IIFA procedure detached dead cells were mechanically removed resulting in a lower number of stained cells at this time point (Figure 2B). HEK-293 cells reached 0.63 MOI (Figure 2C). In contrast, bovine cells had a much lower MOI, in PBK, MDBK, and BEK cells at seven dpi, MOI of 0.14, 0.18, and 0.34 were detected, respectively (Figure 2D–F). Comparing the average of the three human and three bovine MOIs, a significant difference was shown from five dpi (***p*-values 0.0011 and *****p*-values < 0.0001 on five and seven dpi, respectively) (Figure 2I).

### 3.2. Viral Genome Load in Human but Not in Bovine Kidney Cells Increased Significantly with Time

Quantitative real-time PCR of both intra- and extracellular gRNA was measured in triplicate after total RNA extraction from the mentioned infected cells and supernatants (Figure 4). Viral RNA was quantified by targeting the nucleoprotein gene. The intracellular viral RNA in human cell lines revealed an average two-log increase during the infection cycle by five dpi compared to zero dpi (*p* = 0.0087, < 0.0001, 0.0012 in HEK-293, SW-13 and HMC, respectively) and at seven dpi (*p* < 0.0001, < 0.0001, and 0.0180 in HEK-293, SW-13 and HMC, respectively) (Figure 4A). The viral load reached its peak by five to seven dpi with an average of six to seven-log copies/mL (mean 4.8 × 10^7^, 4.0 × 10^7^, and 8.5 × 10^6^ copies/mL in SW-13, HEK-293, and HMC, respectively). At each time point, extracellular viral RNA was lower than intracellular viral RNA, but showed similar increases in both primary and immortalized human cells over time (Figure 4B). Comparing with zero dpi, HEK-293 showed a significant difference at five dpi (*p*-value 0.01 and mean 6.7 × 10^5^ copies/mL) and at seven dpi (*p*-value 0.0005 and mean 9.1 × 10^5^ copies/mL). The SW-13 cell line showed a significant increase in the extracellular genome load from day three p.i (*p*-values 0.0002 < 0.0001, < 0.0001 on three, five, and seven dpi respectively and mean 1.9 × 10^6^, 1.7 × 10^6^, and 1.3 × 10^6^ copies/mL on three, five, and seven dpi, respectively) (Figure 4B). HMC cells also displayed a significant increase at five dpi (*p*-value 0.0258 and mean 5.9 × 10^5^ copies/mL) and at seven dpi (*p*-value < 0.0004 and 4.9 × 10^5^ copies/mL) (Figure 4B).

None of the bovine kidney cells showed a significant increase in viral load during the experiment (Figure 4). Similar to human cells, intracellular viral loads were higher than extracellular in bovine cells, but the differences were not significant (Figure 4A). While the intracellular RNA load increased (and decreased in PBK) by time, extracellular RNA remained on the same level. Among bovine kidney cells, PBK displayed the highest intracellular viral load with a peak at the three dpi (mean 2.5 × 10^6^ copies/mL) and then decreased at seven dpi (mean 2.9 × 10^5^ copies/mL). By the end of seven dpi, MDBK had a mean of 3.9 × 10^5^ and BEK had mean 4.5 × 10^5^ copies/mL, respectively. The extracellular RNA reached a maximum at one dpi and three dpi of mean 2.4 × 10^5^ copies/mL and mean 1.7 × 10^5^ copies/mL in MDBK and BEK, respectively (Figure 4B). PBK displayed an average four-log except at five dpi with a mean of 1.3 × 10^5^ copies/mL. The mean viral load was also measured through three serial passages in each cell line (Figure A1). During the passages, the mean viral load in HEK-293, measured at the end of the first replication cycle, increased from 7.1 x 10^7^ copies/mL to 1.2 × 10^8^ copies/mL by the third passage, SW-13 decreased from mean 3.6 x 10^7^ copies/mL to mean 4.8 x 10^6^ copies/mL, and HMC changed from mean 5.1 x s10^6^ copies/mL to mean 4.0 x 10^5^ copies/mL by the third passage, respectively. During three consecutive passages, the viral load decreased in all bovine cell lines by at least two-log copies/mL. In the third passage, MDBK had mean 1.7 × 10^3^ copies/mL, BEK had mean 6.7 × 10^3^ copies/mL, and PBK had mean 7.5 × 10^4^ copies/mL of CCHFV gRNA. 

### 3.3. Human Kidney Cells Display Higher Cytotoxicity

In the LDH assay, cells were infected with virus at 0.1 MOI and the cytotoxicity levels were measured in the following three days. Although, cytotoxicity increased over time in human and bovine kidney cells, higher values of LDH activity were detected in both immortalized and primary human cell lines as compared to bovine cell lines (Figure 5). In immortalized human cell lines (HEK-293, SW-13) at three dpi the cytotoxicity level reached an average 20.31% and 18.72% in HEK-293 and SW-13 respectively compared to immortalized bovine cells with 3.4% and 7.9% in MDBK and BEK cell respectively. The averageLDH level reached 41.62% in primary human HMC cells compared to 24.80% in the primary bovine PBK cells. Both human and bovine primary kidney cells showed a higher rate of cytotoxicity than the immortalized cells. Using Tukey’s multiple comparison test, we found a significant difference on one dpi between SW-13 and MDBK (*p*-value = 0.0224), on two dpi HEK-293 compared to BEK (*p*-value = 0.0061) and MDBK (*p*-value = 0.0019) and SW-13 compared to BEK (*p*-value = 0.0183) and MDBK (*p*-value = 0.0059). At three dpi significant differences were shown between HMC and PBK (*p*-value = 0.0001), HEK-293 compared to BEK (*p*-value = 0.0035) and MDBK (*p*-value = 0.0001) and SW-13 compared to BEK (*p*-value = 0.0122) and MDBK (*p*-value = 0.0004) (Figure 5).

## 4. Discussion

CCHFV is prioritized for research and vaccine/antiviral development in the WHO R&D Blueprint, due to its epidemic potential [32]. Human cases are usually seasonal and agricultural activity-associated, with large domesticated animals intimately involved in virus maintenance and transmission [33,34]. A wide range of cells originating from invertebrates to vertebrates has so far been demonstrated to be susceptible to CCHFV [35,36,37]. Although it was previously shown that the liver is one of the primary replication sites of CCHFV [14,38,39], the virus has systematically spread to other tissues such as the kidney [20,21,22]. Since human and mammalian kidney cell lines such as human adrenal gland/cortex (SW-13), rhesus monkey kidney (LLC-MK2), or Syrian golden hamster kidney (BHK-21) are among the most commonly used cells for in vitro CCHFV research [16,17,18], in this study we used kidney cells as well. The information on viral growth characteristics in human and animal cells are still scarce. We carried out this study to characterize and compare CCHFV replication in bovine and human primary and immortalized kidney cells. 

Based on the inspection of fluorescent foci in IIFA and CPE production (Figure 1 and Figure 3), CCHFV infection in bovine cell lines displayed a restricted, islet-like formation where infected cells were in close contact with each other in the culture system. It has been previously demonstrated that in mosquito cells, Bunyamwera orthobunyavirus causes filopodia-like structures between cells to facilitate the direct spread to uninfected cells, instead of entering into the extracellular space [40]. Our results suggest that in bovine cells the virus uses a cell-to-cell spreading (CCS) process instead of extracellular spreading. It was shown that the CCS process has a less damaging effect on the cells of the endothelial membrane [25,40]. In our study, the bovine cells showed lower LDH, less CPE, and lower MOI (Figure 1, Figure 2 and Figure 5). We have revealed that the cytotoxicity of CCHFV Ank-2 strain was almost double (24.8% vs. 41.62%) in primary human kidney cells than in primary bovine kidney cells. This data was further confirmed in all immortalized cell lines of human and bovine origin. 

In addition to causing less cellular damage, CCHFV is less productive in bovine cells. To confirm that this proposed mechanism plays a role in different outcomes of infection, we carried out a quantitative analysis of the intra- and extracellular virus load by means of quantitating the viral genome based on the S segment (Figure 4). As we expected, the level of both intra- and extracellular viral load increased over time in human cells. However, in bovine cells, only the intracellular viral load increased. Furthermore, according to the MOIs, the virus infected less than 34% of the bovine kidney cells. The restrictions in cell-free virus spread are usually caused by the host immune system, therefore CCS might contribute to the immune evasion mechanisms [24,41,42,43]. A less damaging and less productive viral replication might cause a more tolerable infection in the bovine host immune system. Cell-free spread of viral particles allows the infection of distant cells, while cell-to-cell transmission leads to local spreading [43], therefore CCS might lead to a slower CCHFV infection. The early-onset of CPE and high IIFA scores in human cells strongly support this hypothesis (Figure 2 and Figure A1). Similar observations were reported for Ebola- (EBOV) and Marburg viruses (MARV), which replicate more rapidly in human-origin HuH7 cells than in bat-origin R06E-J cells [23]. Different cell types might give a different affinity to CCS viral spread. For instance, spreading along neurons is a well-known strategy used by neurotropic viruses [43,44]. Since CCHFV is a neurotropic virus [12,15], CCS might play a role in its replication in the brain and in other organs as well. 

The importance of innate immune responses especially interferon class cytokines in CCHFV pathogenesis was documented by the evidence of lethal disease in IFN deficient (Stat-1–/– or Ifnar–/–) but not in wild type mice [45]. We have observed no significant difference in the quantity of attached viruses in any cell line at zero dpi, suggesting that the tropism variations might be related to different cellular mechanisms and host factors. The species-specific traits in innate immune systems are needed to be considered in follow-up studies. Since there are many prominent differences in human and bovine immune systems [46], these dissimilarities in host resistance mechanisms probably play an important role in the distinct outcomes during virus and host interactions.

Viral load is a very important parameter in the consequence of infections, if not the most relevant clinical prognostic factor for CCHFV infections [47,48,49,50]. In the clinical evaluations, a human viral load exceeding seven to eight-logs is usually associated with death or severe clinical manifestations, while patients with a viral load under six-log usually have higher survival rates [47,51]. In accordance with particular in vivo studies [49,50,52], we have observed that human kidney cells reached virus loads of seven-log during weekly cycles while the maximum load in bovine cells remained under six-log. In weekly measurements, the viral load did not fluctuate significantly from the initial amount of inoculum in permanent or primary bovine kidney cells. Interestingly, the level of intra- and extracellular viral load differed from each other. The reason might be that CCHFV is a cell-associated virus, its glycoproteins localize in the intracellular membrane-containing compartments such as ER and Golgi complex, therefore, the amount of virus is higher inside the cells [53].

The high susceptibility of in vivo human-to-human CCHFV infection has been shown. There are reports suggesting that human-to-human CCHFV transmission is possible through sexual contact [39,54,55] with further evidence from non-human primates [15]. Nosocomial transmission of CCHFV has been reported through blood exposure, injury, and even through aerosol exposure [56,57]. In our experiment, serial passages were made to see how the virus can adapt to different cell lines. During the three serial passages, the viral load in infected human kidney cells fluctuated within a maximum range of one-log, while the mean viral load in bovine cells decreased at least two-log copies/mL (Figure A1). We recorded a continuous increase in the HEK-293 cell line (mean viral load changed from 7.1 × 10^7^ to 1.1 × 10^8^ and then to 1.2 × 10^8^ copies/ mL). The viral load decrease in bovine cells compared to human cells supports our hypothesis of the virus being less adaptive/susceptible to bovine cells.

We found small differences in how primary and immortalized cells responded to CCHFV. The main difference was primary cell lines showed a higher rate of LDH based cytotoxicity, which may be due to primary and immortalized cell differences in chromosomes, cell morphology, and/or cell number/plate. The primary cells from the respective species are from different locations/origins of the kidney (glomerulus vs. cortex), which seemingly did not affect the measured patterns.

To our knowledge, our findings, representing an in vitro comparison of CCHFV infected animal and human kidney cells, is the first study in this field. However, the study has potential limitations, since assessment parameters based on CPE scores and 80% CPE estimation in the serial passages are subjective and prone to observation biases. We are currently investigating differential transcriptional responses in various cell lines to precisely characterize intracellular events associated with CCHFV infections. Finally, the effect of host-specific patterns of virus infection in other known tissue targets (e.g., hepatic) has not yet been investigated. 

In conclusion, the viral spreading mechanism, viral growth kinetics, the speed, and cytotoxicity of CCHFV replication differ in human and bovine cells of kidney origin, as documented by different assays in this study. Correlating with disease susceptibility, human cells seem to be more permissive to CCHFV infection than bovine cells, with marked virus replication and prominent cell pathology.

## Figures and Tables

**Figure 1 viruses-12-00685-f001:**
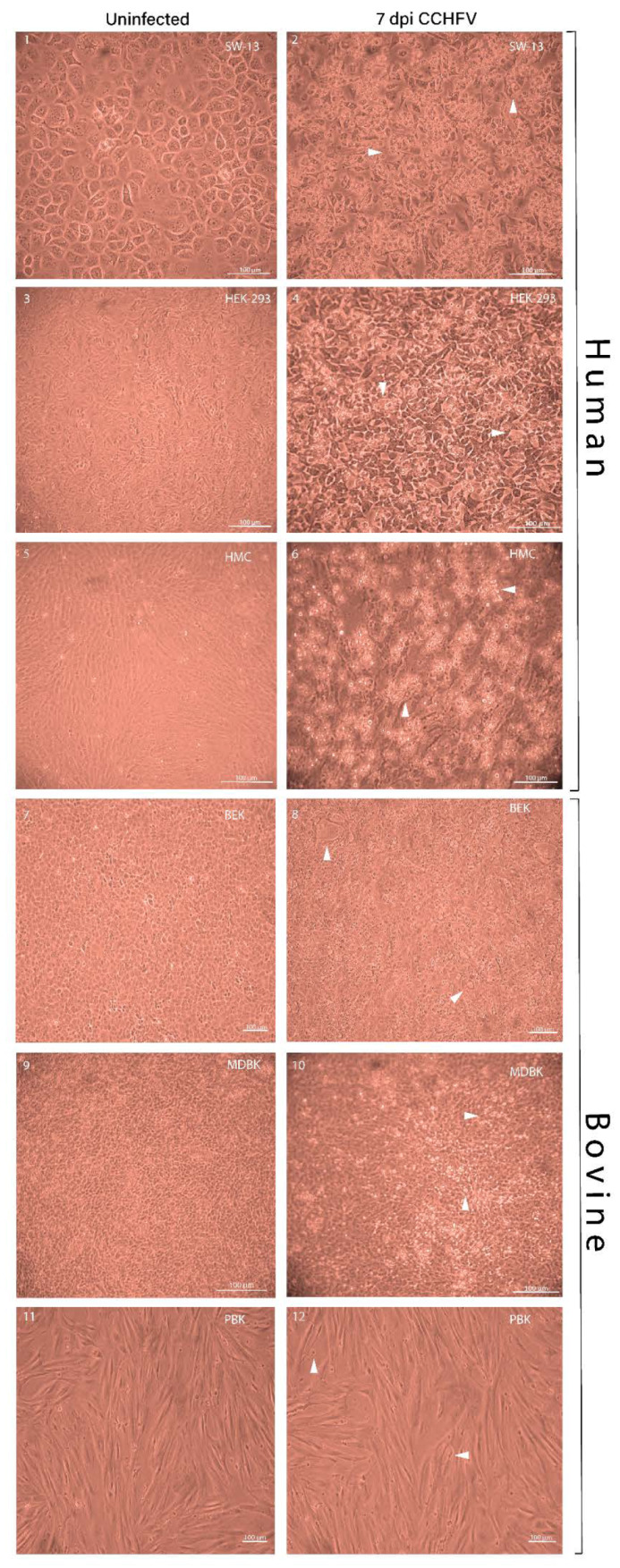
Cytopathic effects in human and bovine kidney cells infected with CCHFV at seven dpi. CPE on primary and immortalized kidney cells were visually scored on one, two, three, five, and seven day-post infection (only day seven shown) by Nikon Eclipse TS-100 inverted light microscope. Seven dpi uninfected cell lines were used as the cell control. Human cells displayed overall cell detachment and cell rounding while bovine cell lines showed localized degeneration and cell deformation. Magnification ×10 (Scale bar: 100 µm). (1) Uninfected SW-13; (2) infected SW-13; (3) uninfected HEK-293; (4) infected HEK-293; (5) uninfected HMC; (6) infected HMC; (7) uninfected BEK; (8) infected BEK; (9) uninfected MDBK; (10) infected MDBK; (11) uninfected PBK; (12) infected PBK. Arrowheads in images indicate pathological changes observed in the respective cell lines.

**Figure 2 viruses-12-00685-f002:**
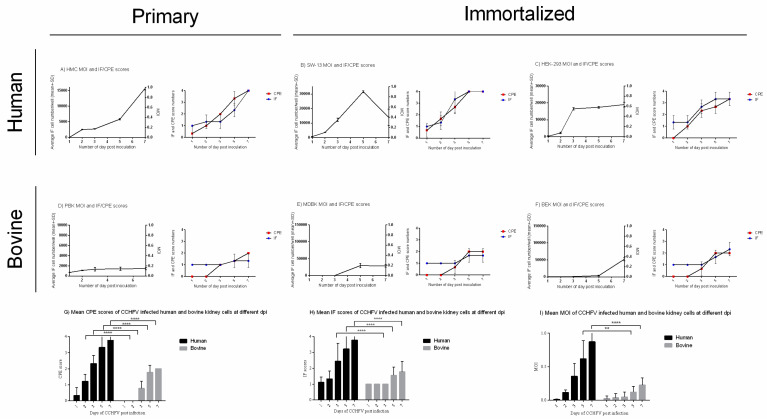
MOI, CPE, and IF staining scores of CCHFV infected human and bovine kidney cells in CCHFV infected cells on days one, two, three, five, and seven dpi. The CPE scoring system was based on assigning numbers (from zero to four) to various levels of CPE where zero indicates no visible change, one indicates CPE up to one-quarter (1%–25%), two up to half (26%–50%), three up to three-quarters, (51%–75%) and four up to all (76%–100%) cells of the monolayers. Uninoculated cells were used as negative controls. The mean percentage of positive cells with green fluorescence emission in 24-well plates measured in triplicates were used to calculate immunofluorescence scores. The grading was based on a one to four scoring system: one indicates up to one-quarter (1%–25%), two indicates up to half (26%–50%), three up to three-quarter (51%–75%) and four up to almost all (76%–100%) cells emitting fluorescence. Uninoculated cells were used as the negative control. The total cell number was counted by Cellcounter. MOI was counted by the following formula: average fluorescent cell/area × 20 area/well × dilution divided by cell number/well (Equation (1)). Mean and SD were measured from triplicate measurements. MOI and CPE-IF scores display major differences between human and bovine cells with higher scores and levels in human cells in all indicated factors. (**A**) HMC MOI, CPE, and IF scores; (**B**) SW-13 MOI, CPE, and IF scores; (**C**) HEK-293 MOI, CPE, and IF scores; (**D**) PBK MOI, CPE, and IF scores; (**E**) MDBK MOI, CPE, and IF scores; (**F**) BEK MOI, CPE, and IF scores. Differences in the mean of three human and the mean of three bovine kidney cells MOI, CPE, and IF scores. (**G**) Mean human and bovine kidney cells CPE scores; (**H**) mean human and bovine kidney cells IF scores; (**I**) mean human and bovine kidney cells MOI scores. The mean and SD were based on triplicate measurements. ** *p*-value ≤ 0.01, **** *p*-value ≤ 0.0001.

**Figure 3 viruses-12-00685-f003:**
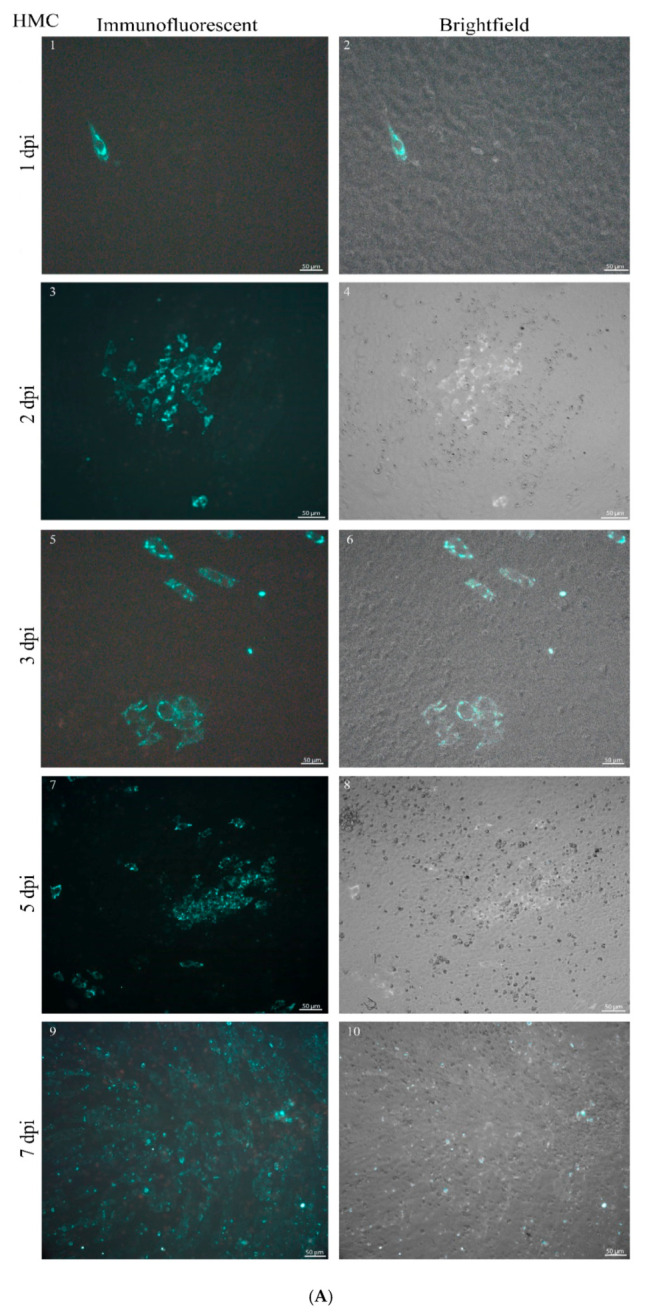
IIFA in primary and immortalized human and bovine kidney cells infected by CCHFV at days one, two, three, five, and seven dpi. Measurements were made in triplicates. Uninoculated cells were used as control. Merged brightfield and fluorescent images are indicated at the right side of the picture. Human kidney cells were infected with a broad distance from each other starting from three dpi, while infected bovine kidney cells remained restricted during the whole infection cycle and bigger islet-like formations were on view from five dpi. Magnification 20x (Scale bar: 50 µm). (**A**) (1,2) HMC at one dpi; (3,4) HMC at two dpi; (5,6) HMC at three dpi; (7,8) HMC at five dpi; (8,10) HMC at seven dpi. (**B**) (1,2) SW-13 at one dpi; (3,4) SW-13 at two dpi; (5,6) SW-13 at three dpi; (7,8) SW-13 at five dpi; (9,10) SW-13 at seven dpi. (**C**) (1,2) HEK-293 at one dpi; (3,4) HEK-293 at two dpi; (5,6) HEK-293 at three dpi; (7,8) HEK-293 at five dpi; (9,10) HEK-293 at seven dpi. For a better overview of the islet-like formations, three different magnifications were used: 5x (scale bar: 200 µm), 10x (scale bar: 100 µm), and 20x (scale bar: 50 µm) indicated in each picture. (**D**) (1,2) PBK at one dpi; (3,4) PBK at two dpi; (5,6) PBK at three dpi; (7,8) PBK at five dpi; (9,10) PBK at seven dpi. (**E**) (1,2) BEK at one dpi; (3,4) BEK at two dpi; (5,6) BEK at three dpi; (7,8) BEK at five dpi; (9,10) BEK at seven dpi. (**F**) (1,2) MDBK at one dpi; (3,4) MDBK at two dpi; (5,6) MDBK at three dpi; (7,8) MDBK at five dpi; (9,10) MDBK at seven dpi.

**Figure 4 viruses-12-00685-f004:**
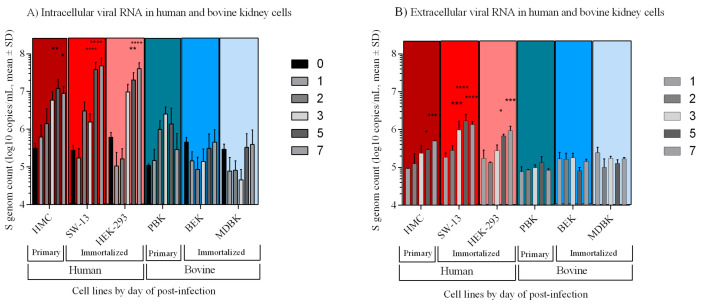
Differential kidney cell line susceptibility to CCHFV, defined by intra- and extracellular gRNA copies at zero, one, two, three, five, and seven dpi. Measurements were taken in triplicate. The results represent both intra- and extracellular viral RNA. The mean viral loads on day one, two, three, five, and seven were compared to the mean viral load at day zero (1 h post-CCHFV inoculation). A significant increase in viral load was measured only in human cell lines. (**A**) Intracellular RNA in immortalized and primary cell lines; (**B**) extracellular RNA in immortalized and primary cell lines. All calculations based on log-transformed viral loads (copies/mL). * *p*-value ≤ 0.05, ** *p*-value ≤ 0.01, *** *p*-value ≤ 0.001, **** *p*-value ≤ 0.0001.

**Figure 5 viruses-12-00685-f005:**
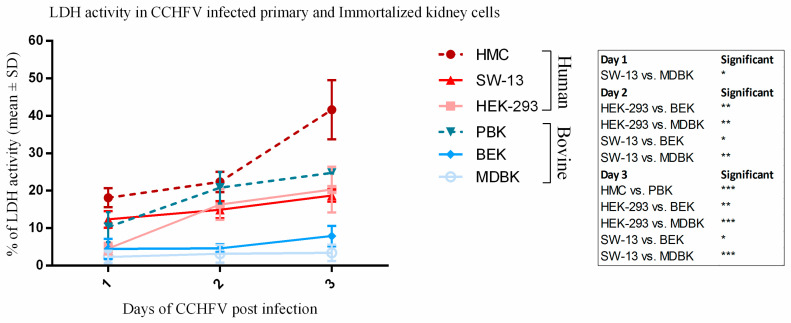
Cell cytotoxicity measured by LDH assay in CCHFV infected immortalized and primary human and bovine kidney cells at different time points. Measurements were taken in triplicate. The mean percentage cytotoxicity was seen at 0.1 MOI of virus infection at one, two and three-days post-CCHFV infection in immortalized kidney cells (BEK and MDBK). Dashed lines indicate primary, continuous lines indicate immortalized cell lines. In immortalized human cell lines at three dpi the cytotoxicity level reached an average of 19.51% compared to an average of 5.65% in immortalized bovine kidney cells and 41.62% in primary human kidney cells compared to 24.80% in the primary bovine kidney cells.

**Table 1 viruses-12-00685-t001:** Human and bovine cell lines used in the present study.

Cell Line	Abbreviation	Culture Requirements	Resources
Human			
Human primary mesangial cell line	HMC	Eagle’s minimum essential medium	Hacettepe University, Ankara, Turkey
Human adrenal gland/cortex Scott and White No. 13	SW-13	Leibovitz’s L-15 medium	National Public Health and Medical Officer Service, Hungary
Human embryonic kidney	HEK-293	Minimum essential medium alpha	Departmental culture collection: ATCC no. CRL-1573
Bovine			
Primary bovine kidney cortex cell	PBK	Eagle’s minimum essential medium	In-house
Madin–Darby bovine kidney	MDBK	Eagle’s minimum essential medium	Departmental culture collection: ATCC no. CCL-22
Bovine embryonic kidney	BEK	Eagle’s minimum essential medium	Dipartimento di Salute Animale, Sezione di Malattie Infettive, Facoltà di Medicina Veterinaria, via del Taglio 8, 43100 Parma, Italy

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
