# Peer review of "Differential Growth Characteristics of Crimean-Congo Hemorrhagic Fever Virus in Kidney Cells of Human and Bovine Origin"

_viruses, 2020, doi:10.3390/v12060685_

Round 1

Reviewer 1 Report

Line 177: " To measure lactate dehydrogenase (LDH) levels of CCHFV in the cells at the first 3 days of 178 infection, we used …".

 What did the authors measure the LDH levels of, the CCHFV or the cells?  If LDH levels of cells were measure, this sentence needs corrected, e.g. "To measure lactate dehydrogenase (LDH) levels of CCHFV-infected cells at the first 3 days of 178 infection, we used...". If the authors measured the LDH levels of the virus, please provide detail information of the method.  

Author Response

Line 179 - CCHFV-infected is added

Reviewer 2 Report

The authors have made several edits and additions which have addressed my questions and comments, and have strengthened the manuscript further.

Author Response

We would like to thank to the Reviewer-2 for his/her valuable contribution to our study. 

Reviewer 3 Report

In this revised version of their paper the authors have addressed the issues I raised during the review of the originally submitted paper.

I do not have additional comments regarding the scientific content of this revised version.

Author Response

Authors that to Reviewer-3 for his/her valuable comments on our manuscript. 

This manuscript is a resubmission of an earlier submission. The following is a list of the peer review reports and author responses from that submission.

Round 1

Reviewer 1 Report

major comments:

1. Line 80-104. One of key topic under the title is difference between primary and immortalized cells. It is critical to clearly separate the three human cells and three bovine cells into primary and immortalized two groups, besides human vs bovine. To separate primary and immortalized cells, a table including resources, key culture requirements, and relevant references will be helpful.

2. Line 175. Please provide clear evidence to support measured (LDH) levels of CCHFV, not cells.

3. Throughout all three result sections, subtitles show comparisons between bovine cells and human cells, but comparing between primary and immortalized cells is also mixed in each section. Since “difference between primary and immortalized cells” is being claimed in the title, the author need separate and make new section to compare the differences between  primary and immortalized cells. Otherwise, the title is overclaiming on the data.

Minor comments:

1. Line 160. Does Alexa Fluor 488 protein G conjugate (e.g. P11065, ThermoFisher) belong to an antibody? The methodological mechanism for this reagent is "...Protein G is a bacterial protein similar to protein A but with superior selectivity for mouse and human IgG....".

2. Line 195, spell out SD

Reviewer 2 Report

General comments:

In “Differential growth characteristics of Crimean-Congo Hemorrhagic Fever Virus in primary and immortalized kidney cells of human and bovine origin” Földes et al. examine the differences in in vitro susceptibility and replication patterns of CCHFV in both bovine cells and human cells. They report on differences in viral spreading mechanisms, viral growth kinetics, and the speed and cytotoxicity of viral replication in the different cell lines. Their conclusions correlate the increased permissiveness to CCHFV observed in human cells compared to bovine cells to the variances observed between humans and bovine species in disease susceptibility.

Main comments:

The data presented are interesting, and the experimental methods used to examine the difference between cells lines of the two origins well thought out. I also appreciate the very well written and, most importantly, thorough material and methods section. Indeed, as a whole, the manuscript is very well written.

The questions the authors are asking are very valid –  trying to determine the differences in kinetics between two cell lines. My main comment would be that there are several clear experiments that should be performed that aren’t presented here. Firstly, simple growth curve experiments of either high or low MOI (or ideally both) should be performed to examine the amount of infectious virus that is produced by each cell line. While looking at intra- and extra-cellular RNA is interesting, this is very hard to correlate to infectious virus titers without specifically designed assays and correct controls. I think it would be appropriate here, if the purpose of the paper is to examine CCHFV growth characteristics in vitro, to see how titers of infectious virus vary between the cells line throughout a growth experiment. Secondly, western blot analysis of viral protein expression throughout these growth curves would also give valuable data on the intracellular dynamics of viral transcription and translation within these different cells line. Having these data included would greatly strengthen the manuscript.

I am also a bit confused by the process of generating the “average IF – MOI “graphs in figure 2. Firstly, the the figure’s current format the fonts used on the axes are too small and should be revised in the final version. Secondly, if I have this correct, the authors have used IF data to calculate the effective MOI at each time point based on cell density after infection? If the intent of the authors is to show the difference in susceptibility between human and bovine-origin cells then I would maybe suggest infecting in the cells at a standard MOI calculated prior to infection and then plotting the relative (over uninfected cells) IF data at each time point? Also, are the data from the cells imaged in Figure 3 those used to generate the data in Figure 2? Because it seems as if the fluorescence levels for the MDBK cells at 5dpi are much higher than the values shown in Figure 2 would suggest?

In the LDH assay, were the values used relative to uninfected cells at each time point or just raw values? I think this data should definitely be presented as relative to uninfected at each of the three time points assessed if it isn’t already.

Minor comments:

Line 217 – Figure 1: Are the uninfected cells from 7dpi also? This should be made clear in the figure or legend.

Line 284 – Figure 3: The “brightfield” images are merged brightfield + fluorescent, not brightfield. This should be labeled for clarity.

Line 284 – Figure 3 legend. There is a lot of redundancy in this legend, it would benefit from being rewritten to simplify.

Line 298 – Section 3.2. The values for the copies/mL in this section (and others throughout the manuscript) need to have the log number in superscript (or equivalent) i.e. 5 × 107 or 5 × 10e7, not 5 × 107

Line 331 – Figure 4: This figure would benefit from being larger. The variances may also be clearer to the reader if the v-axis started  at 104 for example?

Reviewer 3 Report

General comment:

Crimean-Congo hemorrhagic fever virus (CCHFV), a member of the orthonairovirus genus in the family Bunyaviridae, is a tick-borne virus that can cause severe severe clinical symptoms in humans and high fatality rates of up to 50% in hospitalized patients. In contrast, CCHFV causes limited or none clinical symptoms in wild or domestic animals. Despite CCHFV poses a significant risk to public health, there are currently no vaccines or specific therapies against CCHFV. The mechanisms underlying the striking differences of the pathogenic manifestations of CCHFV infection between human and its natural hosts remain poorly understood. Hence, the significant of the present work by Földes and colleagues examining for the first-time potential factors contributing to the different outcomes of CCHFV infection between human and bovine immortalized and primary kidney and adrenal gland cell lines.

The authors have presented convincing evidence that multiplication of CCHFV occurs to lesser levels in bovine than human cells, which correlated to a lower level of CCHFV induced cell damage in bovine than human cells.  In addition, the authors have presented evidence suggesting that CCHFV spreads mainly cell-to-cell in bovine, whereas in human cells CCHFV spreads involves virus release from infected cells and infection de novo of cells with extracellular virus. The authors proposed that the cell-to-cell mechanism of viral spread results in lower levels of virus induced cell damage, which is supported by lower levels of released LDH and less cytopathic effect in CCHFV infected bovine cells compared to infected human cells.

The paper is clearly written, and the experiments were appropriately performed. Likewise, the results have been clearly presented. A limitation of this work is the lack studies aimed at identifying potential cellular factors that lead to different outcomes of infection between human and bovine cells. In addition, the authors need to provide additional information to clarify several issues in the present paper (see specific comments).  

Specific comments:

Figure 1: The resolution of each picture should be increased. It is very difficult to appreciate the pathological changes indicated by the arrow heads.

Figure 4: Quantification of extracellular RNA is missing values of 0 days pi.

Figure 5: No statistical significance between human and bovine cells? It would be good to indicate the levels of significance on the graph like it was done in figure 4.
